# Bridging Cyanobacteria to Neurodegenerative Diseases: A New Potential Source of Bioactive Compounds against Alzheimer’s Disease

**DOI:** 10.3390/md19060343

**Published:** 2021-06-16

**Authors:** Andrea Castaneda, Ricardo Ferraz, Mónica Vieira, Isabel Cardoso, Vitor Vasconcelos, Rosário Martins

**Affiliations:** 1CISA, Health and Environment Research Centre, School of Health, Polytechnic Institute of Porto (ESS/P.PORTO), Rua Dr. António Bernardino de Almeida, 400, 4200-072 Porto, Portugal; andrea.castanedam98@gmail.com (A.C.); rferraz@ess.ipp.pt (R.F.); mav@ess.ipp.pt (M.V.); 2LAQV-REQUIMTE, Departamento de Química e Bioquímica Faculdade de Ciências, Universidade do Porto, R. do Campo Alegre, 4169-007 Porto, Portugal; 3I3S Instituto de Investigação e Inovação em Saúde, Universidade do Porto, Rua Alfredo Allen, 208, 4200-135 Porto, Portugal; icardoso@ibmc.up.pt; 4IBMC, Instituto de Biologia Molecular e Celular, Universidade do Porto, Rua Alfredo Allen, 208, 4200-135 Porto, Portugal; 5ICBAS, Instituto de Ciências Biomédicas Abel Salazar, Universidade do Porto, R. Jorge de Viterbo Ferreira, 228, 4050-313 Porto, Portugal; 6CIIMAR/CIMAR, Interdisciplinary Centre of Marine and Environmental Research, University of Porto, Terminal de Cruzeiros do Porto de Leixões, Av. General Norton de Matos s/n, 4450-208 Matosinhos, Portugal; vmvascon@fc.up.pt; 7FCUP, Department of Biology, Faculty of Sciences, University of Porto, Rua do Campo Alegre, Edifício FC4, 4169-007 Porto, Portugal

**Keywords:** cyanobacteria, Alzheimer’s disease, natural products, new therapies

## Abstract

Neurodegenerative diseases (NDs) represent a drawback in society given the ageing population. Dementias are the most prevalent NDs, with Alzheimer’s disease (AD) representing around 70% of all cases. The current pharmaceuticals for AD are symptomatic and with no effects on the progression of the disease. Thus, research on molecules with therapeutic relevance has become a major focus for the scientific community. Cyanobacteria are a group of photosynthetic prokaryotes rich in biomolecules with confirmed activity in pathologies such as cancer, and with feasible potential in NDs such as AD. In this review, we aimed to compile the research works focused in the anti-AD potential of cyanobacteria, namely regarding the inhibition of the enzyme β-secretase (BACE1) as a fundamental enzyme in the generation of β-amyloid (Aβ), the inhibition of the enzyme acetylcholinesterase (AChE) lead to an increase in the availability of the neurotransmitter acetylcholine in the synaptic cleft and the antioxidant and anti-inflammatory effects, as phenomena associated with neurodegeneration mechanisms.

## 1. General Introduction

### 1.1. Neurodegenerative Diseases and Natural Compounds

Neurodegenerative diseases (NDs) are highly debilitating conditions that involve the progressive degeneration and/or loss of nerve cells [1]. Despite years of research in the field, the exact cause of the neurodegenerative process is unknown, but it is assumed that results from interrelated mechanisms that trigger neurons degeneration and death—such as synaptic loss, misfolded proteins, reactive oxygen species (ROS), reactive nitrogen species (RNS), and electrophysiological abnormalities [2].

Alzheimer’s Disease (AD) is a NDs and the most common dementia [3]. Cognitive impairment derived from AD interferes with daily life activities, ultimately causing dependence, disability, and even death [4]. Despite efforts to find therapies, currently, there are no available drugs capable of halting the disease progression but only able to mitigate the symptoms. Thus, the search for new drugs has been unceasing, namely the search for compounds of natural origin.

Living organisms represent a rich source of natural compounds, both with pharmacological potential or as lead compounds [5]. Natural products (NP) have been crucial for the development of new drugs applied in various fields of biotechnology, with an ever-growing relevance in the pharmaceutic industry, namely in inflammation, cancer, and NDs [6,7].

Although plants and invertebrates, have been the major sources of NP [8] the microscopic world has also proved to be a prolific source [9]. Even compounds initially attributed to invertebrates were later identified as being produced by symbiotic microorganisms. Two well-known examples are the Brentuximab Vedotin (Adcetris^®^) and the Trabectedin (Yondelis^®^), approved for the treatment of Hodgkin’s lymphoma and the Trabectedin (Yondelis^®^) approved for the treatment of soft tissue sarcoma and ovarian cancer, respectively [10,11]. The first was initially found to be produced by a marine sponge and lately by a symbiotic cyanobacteria, while the second example was firstly attributed to a tunicate and lately to a symbiotic bacterium [12]. Both examples refer to compounds produced by marine organisms. The marine environment has, in fact, emerged as a promising source for novel bioactive NP namely in AD drug discovery [7]. Examples are ziconotide, an alternative for opioids extracted from the cone snail *Conus magus* [13]; and bryostatin-1 from *Bugulaneritina* [14], which has pharmacologic potential for different diseases, including AD [14,15,16].

### 1.2. Cyanobacteria

Cyanobacteria are a morphologically and physiologically diverse group of photoautotrophic organisms [17]. Their early appearance in the history of life enabled them to spread widely, including in environments with extreme conditions such as very high temperatures and salinity. Cyanobacteria high rate of adaptation implies a diversity of survival mechanisms, that include the production of secondary metabolites attractive for biotechnology [18]. As examples, cyanobacteria produce exopolysaccharides (EPS) in order to survive desiccation [19]; trehalose as a mechanism against freeze stress [20]; microsporine-like amino acids (MAAs) and scytonemin (SCY), that act as sunscreens [21,22]; and even protective stress proteins [23]. Other interesting compounds include the phycobiliproteins phycocyanin (PC), phycoerythrin (PE) and allophycocyanin (APC), carotenoids and phenols. C-phycocyanin (C-PC) was found to induce hepatoprotective, anti-inflammatory and inhibitory effects on cytochrome oxidase 2 [24,25]; PE was found to potentialize anti-inflammatory activity, while carotenoids exhibit antioxidant and anti-inflammatory activity, with the ability to lower the risk of heart disease and stimulate the immune system [26]. Considering the nervous system, NP from cyanobacteria have a wide spectrum of action, from toxic molecules capable of neurotoxic activity, such as anatoxin-a(s) [27], to compounds with therapeutic potential for AD such as tasiamide B, an aspartate protease inhibitor of β-secretase, (BACE1) [28].

### 1.3. Objectives of the Review

The severity of the individual, familiar, and societal problems underlying AD and the lack of treatments have driven the constant search for new therapies. NP continue to show increasing importance in the field of pharmacology. Considering cyanobacteria as an interesting group of microorganisms for NP with pharmacological potential, this review aims to compile the existing research that links cyanobacteria to AD in order to contribute to ameliorating the disease in an innovative away.

## 2. Alzheimer Disease

AD is a ND that results from a complex network of mechanisms and factors, including genetics, environment, and lifestyle [29]. In AD, a gradual and progressive deterioration of neurons of the central nervous system (CNS) occurs [30,31], affecting cognition and memory. As previously referred, AD is the most prevalent NDs, covering approximately 60–70% of the dementia cases. Nowadays, nearly 47.5 million (14%) people suffer from this neurological disorder. As life expectancy continues to increase, this number is estimated to increase about 7.7 [32], thus expected to triple to 152 million by 2050 [3]. Prevalence in Europe shows a notable age-related increase from 65 years old, with increasing prevalence values of 0.97%, 7.66% and 22.53% for 65–74, 74–84, and up to 85 years old, respectively [33].

The histological hallmarks of AD include the presence, in the brain, of extracellular amyloid plaques composed by misfolded amyloid-β proteins (Aβ), and intracellular aggregations of neurofibrillary tangles (NTFs), constituted by hyperphosphorylated tau protein [34,35]. It is thought that Aβ plaques develop initially in basal, temporal, and orbitofrontal neocortex regions of the brain to later spread to the neocortex, hippocampus, amygdala, diencephalon, limbic system, and basal ganglia [36]. The structural and functional degeneration seems to be focused on the neocortex and hippocampus which can also be developed by synaptic loss [1].

### 2.1. Alzheimer Disease Hypothesis and Main Therapeutical Targets

Despite the extensive research focus on AD, its origin, development, and progression are still unclear. However, several hypotheses are pointed out that try to explain its development.

#### 2.1.1. Amyloid Cascade Hypothesis (ACH)

In ACH, the Aβ peptide plays a main role on the pathology. It is produced through the sequential cleavage of the transmembrane amyloid precursor protein (APP) by BACE1 [28], followed by γ-secretase. BACE1 cleaves the extracellular domain of APP, releasing the βAPP from the cell surface and leaving the C-terminal of APP (CTF-β), resulting in Aβ peptides. Excessive Aβ peptide suffers a chain of self-assembly steps that culminates with the formation of insoluble Aβ aggregates, deposited extracellularly near to synapses. Aβ aggregates trigger a downstream cascade of events—leading to synaptic and mitochondria dysfunction, endoplasmatic reticulum stress, oxidative stress, DNA damage, neuroinflammation [31,37,38,39]—impairing cognitive function and ultimately death. Aβ plaques-induced neuronal death is also due to alterations in the cellular membrane integrity (formation of pores) and fluidity [29]. Aβ oligomers can also cause abnormal kinases and phosphatases activities, leading to the hyperphosphorylation of tau protein and subsequent NFT formation [34]. Aβ elevated levels and aggregation can alone increase Aβ production [40]. Further contribution for both, Aβ and tau pathology, in AD is provided by the reduced expression of the brain-derived neurotrophic factor (BDNF), which also has a role on memory besides the neuronal growth function [41]

Based on the ACH hypothesis, therapeutic approaches have focused on the reduction of Aβ production, aggregation, accumulation, and on enhancing Aβ clearance. A promising approach is the inhibition of BACE1 [35]. In contrast with γ-secretase inhibitors, the BACE1 inhibitors show higher substrate specificity and fewer effects [42]. However, despite the number of discovered BACE1 inhibitors, several phase III clinical trials have been abandoned due to safety reasons, or because the end-points were not reached, or even due to cognition worsening [43].

Another attractive therapy for AD targeting the Aβ pathology is anti-Aβ passive immunotherapy with antibody administration, based on the peripheral sink hypothesis mechanism [44]. This hypothesis suggests that Aβ peripheral immunological clearance promotes CNS Aβ release through a concentration gradient mechanism [44,45]. To reduce the Aβ-induced synaptotoxicity, the inhibition of the non-receptor Fyn tyrosine kinase is a promising target, which is activated through Aβ oligomers-stimulated cellular prion protein (PrPC), interfering with long-term potentiation. In addition, Fyn seems to correlate with abnormally phosphorylated tau [46].

#### 2.1.2. Cholinergic Hypothesis (CH)

The CH is based on a deficient cholinergic system signaling by acetylcholine (ACh) [47]. Cholinergic neurons are involved in cognitive processes as memory, attention, learning and regulation of the sleep cycle function [48]. In AD patients, the cholinergic neurons that innervate the hippocampus and neocortex are degenerated, which leads to acetylcholine deficiency and loss of cholinergic signaling. In the later phases of AD, butyrilcholinesterase (BChE), which also degrades ACh, increases up to 90% [49,50]. In addition, the choline acetyltransferase (ChAT) involved in ACh synthesis, is decreased [51]. In late stages, both enzymes involved in the cholinergic signaling (synthesis and degradation) have abnormal activities leading to a failure in the cholinergic system [14,52].

AD therapy strategies include the increase of the acetylcholine levels through the inhibition of the AChE, the use of acetylcholine analogues and allosteric modulators of acetylcholine receptors [53,54,55,56,57,58,59]. Among those, the inhibition of AChE has been the most relevant strategy, which includes most of the current treatments [60].

#### 2.1.3. Glutaminergic Hypothesis (GH)

The GH is focused on the crucial role of the inappropriate GluN2A-containing N-methyl-D-aspartate receptors (NMDARs). This hypothesis attributes a relevant contribution of inappropriate stimulation of glutamate receptors on the deterioration of synaptic function. Overstimulation leads to CNS complications and may be related with Aβ plaques interaction, leading to synaptic impairment and neurodegeneration [6,27]. NMDARs are involved in synaptic transmission and synaptic plasticity, (closely related with memory and learning) and its hyperexcitation results in excitotoxicity mediated by increased Ca^2+^ influx, which affects synaptic signaling and function ultimately leading to neuronal death [61,62]. On the other hand, Aβ plaques can overstimulate NMDA receptors with consequent desensitization and higher Ca^2+^ influx [63,64,65].

This glutamate-induced excitotoxicity can be addressed by blocking its NMDAR with NMDAR inhibitors, such as galantamine or through the regulation of Ca^2+^ levels, potentially achieved by NSAIDs such as diclofenac and rofecoxib, which can combat neuroinflammation and regulate tau phosphorylation, through the Rho-GTPases pathway [29].

#### 2.1.4. Tau Hypothesis (TH)

In the TH, the hyperphosphorylation of microtubule-associated protein tau occurs, followed by its aggregation in NTFs [66]. Tau protein is mainly found combined with microtubules in neuronal axons of the brain [67]. Its role in cytoplasmatic transport enables to maintain synaptic function and structure at the same time that regulates neuronal signaling [68,69]. In AD, tau protein is present in hyperphosphorylated and abnormally cleaved forms and conformations. As consequence, tau protein is prone to aggregation and subsequent production of NFT [70]. Hyperphosphorylation of tau causes defective microtubule functioning due to the loss of tubulin polymerization capacity [71,72]. Intracellular NFTs produce neurotoxicity and reduce the number of synapses, deteriorating cell functioning [73]. Moreover, tau has the capacity to spread pathological tau in a prion-like mechanism and can have implications in neurodegeneration and cognition [74]. Additionally, the hyperphosphorylation levels of tau positively correlate with the severity of AD [75].

In the light of this approach is the development of inhibition mechanisms for kinases and tau aggregation, immunotherapeutic agents, and stabilizers of microtubules. Evidence and proposal of a prion-like transmission mechanisms for tau and Aβ can be considered valid approaches for AD treatments [42]. In order to prevent hyperphosphorylation of tau protein, a promising approach is the inhibition of glycogen synthetase kinase-3β (GSK-3β), whose abnormal activity leads to hyperphosphorylation and accumulation of tau protein and even mediation for Aβ plaques production [46,76].

#### 2.1.5. Inflammation and Oxidative Stress Hypothesis

Unlike the rest of the body, the brain immune system is not constituted by peripherical immune cells, instead, the immunological function is exerted by microglia. When microglia are activated into a pro-inflammatory state, neuroinflammation is triggered [77]. The release of cytokines and reactive oxygen species (ROS), as part of the pro-inflammatory function can cause severe neuronal damage when it is unbalanced with the cell repair function, leading to synaptic loss and ultimately neuronal death [77].

As early events on AD, chronic inflammation has been registered [40]. Inflammation seems to be a result of the release of inflammatory cytokines from excessive deposition of microglial cells [29]. In fact, when small Aβ plaques begin to form, the activation and attraction of microglia occur [35]. The excess of Aβ plaques and the constant activation of glial cells provokes the release of inflammatory cytokines, producing neuroinflammation and synaptic loss. Additionally, the tau-containing tangles can also stimulate a neuronal immunological response [77]. Furthermore, cytokines can raise the expression of the insulin-degrading enzyme neuronal apoptosis as well [29]. Therefore, neuroinflammation can be considered a relevant therapeutic direction [29].

Still on an anti-inflammatory approach, the inhibition of the receptor-interacting serine/threonine-protein kinase 1 (RIPK1) has been proposed for the suppression of microglia. However, the selectivity of these inhibitors is a prior concern. With a remarkable entrance to phase I clinical trials for AD treatment, DNL-747 can be the first in-human utilization of RIPK1 inhibitors [44].

Oxidative stress leads to oxidative damage in biomolecules such as DNA, lipids, and proteins. As the brain energy uptake is indeed elevated, this organ is vulnerable to a higher ROS production. The confirmation of high levels of oxidative damage markers and low levels of antioxidants in AD brains suggests that oxidative stress is implicated in AD pathology [40].

Oxidative stress can trigger a cascade of events that leads to neuronal death. It can cause glycose dysmetabolization by damaging glycolytic enzymes and enzymes involved in energy production that results in the loss of ion gradient and calcium dyshomeostasis [78,79]. High Ca^2+^ levels can (1) stimulate endonucleases, phospholipase and proteinase activities, with consequent cell dysfunction; (2) lead to loss of microtubule assembly, which compromise energy transport to/for synapses [80,81]; (3) induce swelling of the cell, leading to alterations on mitochondrial permeability and release of cytochrome c and apoptosis inducing factor 1 [82]; and (4) promote synaptic dysfunction [79].

Oxidative stress can also affect directly nuclear and mitochondrial DNA, impairing all the processes of protein synthesis, including those involved in energy production [83,84,85]. Then, mitochondria dysfunction in AD ATP production and leads to the loss of ion gradient and consequent loss of neurotransmission membrane potential and further synaptic loss [78,86,87].

### 2.2. Current Therapeutics

Currently, anti-AD drugs include AChE inhibitors and an NMDA receptor antagonist. Galantamine, rivastigmine, donepezil, and memantine are the most common drugs and with the exception of memantine, they all are AChE inhibitors [88].

Galantamine, besides its AChE inhibition action, also interacts with nicotinic cholinergic receptors [89], being effective to treat cognition-relative symptoms [30]. It is a competitive inhibitor from a natural source [8], approved in 2001 by FDA under the name of Reminyl [90]. It is appropriate for mild to severe AD [91]. Donepezil is also an inhibitor of AChE, and in addition it interferes on various aspects of glutamate-induced excitotoxicity, reduces the early expression of inflammatory cytokines, thereby acting in several of AD pathogenic stages [30]. It is a non-competitive reversible inhibition with low potency against BChE, with the ability to cross the blood–brain barrier (BBB) and provides a longer and more selective action (compared with previous AChE inhibitors) with easier to manage side effects [27]. Approved in 1996 [92], it is adequate for mild to severe AD [91]. Rivastigmine exerts its reversible inhibitor action on both AChE and BChE, based on a carbamate moiety with higher affinity to AChE than the carbamate moiety of ACh, enabling temporal inactivation of the enzyme [30]. It is a synthetic derivate of the natural compound physostigmine with capacity of crossing the BBB [27]. Rivastigmine was approved in 2000 for mild to moderate AD [91,93]. Finally, memantine is a blocking agent against NMDA receptors, revealing protective properties against NMDAR-mediated excitotoxicity with a strong voltage-dependency. Memantine protects the memory and learning process by allowing “the transmission of transient physiological signals” and induces an additional neuroprotective effect through the stimulation of neurotrophic factor release from astroglia [91]. Memantine was approved in 2003 by the FDA for moderate to severe AD [94,95].

There has been an investment on the discovery of new AChE inhibitors from a wide range of source and molecules, among which flavonoids can be a great target for potent AChE inhibitors in addition to its antioxidant properties, as the flavonoid Galagin from the rhizome *Alpiniae officinarum* [30]. Cyanobacteria are another potential AChE inhibitor source further discussed on this review.

## 3. Cyanobacteria Potential in Alzheimer Disease

With a focus on AD, cyanobacteria have shown an increasing therapeutic potential and a feasible source of novel active drugs. The history behind the relevance of cyanobacteria in neuroprotection comes from the production of neuroactive substances such as anatoxin-a(s), microcystins, and nodularin, which exert a competitive advance in grazing defense, by reducing palatability and avoid predation [96]. Particularly regarding the cholinergic system, AChE inhibitors in cyanobacteria seems to be involved in inhibiting the colonization of colonies and filaments by other organisms, since AChE inhibitors were found to inhibit invertebrate larval settlement [97].

Examples of bioactivity against AD from cyanobacteria extracts or compounds are listed in Table 1 and summarized in Table 2.

The *Symploca* sp. compound, tasiamide B (Figure 1), has been identified as a potential BACE1 inhibitor [98,99]. Also, tasiamide F (Figure 1), an analogue of tasiamide B, isolated from *Lyngbya* sp. was found to inhibit BACE 1, however with 8 fold less effectivity. This low BACE1 inhibitory potential of tasiamide F compared with tasiamide B was attributed to minor alteration on residues that have hydrophobic interactions with the receptors pocket, which exert the inhibitory effect [100]. Also, series of tasiamide B derivatives were assessed for their ability to inhibit the activity of BACE1. Results indicate that tasiamide B is a good template for the development of selective BACE1 inhibitors [101].

The search for AChE inhibitors in cyanobacteria has also been one of the approaches followed. Anatoxin-a(s) (Figure 2) found in several cyanobacteria, and initially extracted from *Anabaena flos-aquae*, has been largely studied for AChE inhibitory potential. Results on AChE inhibitory assays showed that anatoxin-a(s) was able to inhibit both AChE and BChE, with more specificity towards AChE. In in vivo experiments confirmed these results as rats treated with the toxin showed similar signs of anticholinesterase poisoning [102].

Several methanolic extracts from the São Paulo Botanical Institute Cyanobacterial Culture Collection—*Calothrix* sp. CCIBt 3320, *Tolypothrix* sp. CCIBt 3321, *Phormidium* cf. *amoenum* CCIBt 3412, *Phormidium* sp. CCIBt 3265, and *Geitlerinema splendidum* CCIBt 3223 exhibit reversible inhibition of AChE in in vitro studies. Furthermore, in vivo studies in mice showed systemic effects similar to those observed with anticholinesterase treatments [103].

Nostocaroline (Figure 2) isolated from the strain *Nostoc* 78-12A was found to be a potent BChE inhibitor with a IC_50_ value of 13.2 µM, comparable to galantamine [104]. In addition to this inhibitory activity, Nostoc strains such as *Nostoc elispsoporum* CCAP 1453 were studied for their antioxidant capacities on Trolox equivalents [105]. Their hexane, ethyl, and water fractions showed high values of Trolox equivalents, varying from 2.37 +/− 1.15 to 21.09 +/− 1.83 µmol Trolox/g according to the *Nostoc* strain [105].

Phycobiliproteins from cyanobacteria were found to be attractive for AD therapy and prevention, namely PC and PE. Through molecular docking, an energetically favorable interaction between PC from *Leptolyngbya* sp. and BACE1 was found using the nematode AD-model *Caenorhabditis elegans*. The crystal structure and interaction of PC with BACE1 revealed a good interaction between both compounds [106]. Moreover, PE, also from *Leptolyngbya* sp., was found to inhibit BACE1 in in vitro assays. The in vitro high affinity between PE and BACE1 were further confirmed with in vivo studies with *C. elegans* worms, where a reduction on Aβ deposition was registered. These results confirm PE as a potential active principle for development of new drugs against AD [107].

The cyanobacteria genus *Spirulina* has been one of the most studied concerning anti-AD treatments, mainly regarding its antioxidant, anti-inflammatory, and neuroprotective effects. C-Phycocyanin (C-PC) from *Spirulina* sp. showed antioxidant, anti-inflammatory, and neuroprotective activities. C-PC was found to inhibit Aβ_40/42_ fibril formation [108]. When male Wistar rats exposed to the neurotoxic agent Tributyltin chloride (TBTC) were treated with C-PC, a remarkably reduction on ROS generation was registered [109]. On brain homogenates, the protein carbonylation and the lipid peroxidation increased by TBTC were efficiently restored to control values; the enzymatic activities related with the antioxidant response altered by TBTC were also restored to control levels [109]. Apart from its antioxidant effect, C-PC also showed an effect on inflammatory molecules by combating the TBTC induced NF-kB p65 upregulation and caspase-12 activation [109]. The same study also revealed the efficacy of C-PC at reverting the nefast effects of TBTC in microglia, oligodendroglial, and astroglia populations [109].

From *Spirulina platensis*, a water extract rich in C-PC, administered in the dietary supplementation of transgenic SAMP8 mice, enhanced the results of passive and active avoidance behavioral tests, to similar values of those of the external control group SAMR mice, suggesting that *S. plantesis* inclusion into the diet can improve emotional memory [110]. Through the analysis of molecular markers, the effects on memory could be explained by the lower brain and hippocampal Aβ accumulation, and by the enhanced brain redox status (lipid peroxidation, CAT, SOD, and GSHx activities) observed on supplemented mice compared with control groups [110]. Similarly, *S. plantesis* daily administration by oral intubation during 24 days on male SW mice, induced neuroprotectant effects against the neuronal damage induced by kainic acid (KA) administration [111]. Although *S. plantesis* did not protect against KA-induced seizures, it reduced KA-mortality by 40%, partially protected against neuronal cell death and reduced the number of atrophic neurons in CA3 hippocampal region [111].

Concerning PE, this compound isolated from *Phormidium* sp. A09DM presented neuroprotective effects on *C. elegans* and *Drosophila melanogaster* due to its antioxidant effect. In fact, PE reduced ROS levels in *C. elegans* induced with oxidative stress and reduced oxidative stress by 14.5% in the AD phenotype *C. elegans* CL4176. In addition, on *D. melanogaster* under oxidative stress, PE improved locomotion, and enhanced antioxidant enzymes activities [112]. Also, from *Phormidium* sp. A09DM the phycobiliprotein APC increased stress tolerance in *C. elegans* CL4176 by showing higher ROS scavenging activity, attenuated Aβ aggregation and increased the lifespan of the worms [113].

Miranda et al. (1998) [114] studied, in vivo, the effects of a 5 g daily dose of *Spirulina maxima* extract on Wistar rats, in plasma and in the liver. While liver antioxidant capacity was similar after the *S. maxima* treatment, the plasma antioxidant capacity increased up to 71% after 7 weeks of treatment. The same study defined, on an in vitro experiment, a *S. maxima* extract IC_50_ value of 0.18 mg/mL for the reduction of oxidation [114]. Other studies showed the ability of a *S. maxima* ethanolic extract to increase the concentration of brain-derived neurotrophic factor (BNDF) during Aβ_1-42_-induced neurotoxicity in PC12 cells, preventing Aβ-induced neuronal death [115]. The growth of *S. maxima* on Zarrouk’s medium NaNO_3_ and/or combined with phenylalanine (L-PA) had positive effects on the production of total phenolics and flavonoids, which in turn resulted in the DPPH radical scavenging activity and antioxidant effects towards CCI4-induced lipid peroxidation in liver homogenate [116]. Also from a review on *S. maxima* phycocyanobilin, it was suggested that this compound could modulate the microglia cytotoxicity and neuronal function and survival through its NADPH oxidase inhibitory activity [117].

**Table 1 marinedrugs-19-00343-t001:** Compilation of cyanobacterial bioactivity against AD pathology.

Genera/Specie	Compound/Extract	Mechanism/Effect	In Vitro Assays	In Vivo Assays	Reference
*Symploca* sp.	Tasiamide B	BACE1 inhibition	BACE1 inhibition assay H4 cells: HPLAP-APP Reporter Assay; CTF analysis; BBB PAMPA; CHO 2B7 cells: Secreted Aβ Assay.	CF-1 Mice: plasma and brain Aβ levels by anti-Aβ antibody; plasma and brain compounds stability monitoring by HPLC	[98,99]
*Leptolyngbya* sp.	Phycocyanin	BACE1 inhibition	In silico assay of molecular docking	Transgenic *Caenorhabditis elegans* AD-model	[106]
	Phycoerythrin	BACE1 inhibition	Thermodynamics of binding using surface plasmon resonance (SPR); isothermal titration calorimetry (ITC); enzyme activity by kinetic parameters.	*Caenorhabditis elegans* CL4176 transgenic AD model worm: Aβ reduction by Thioflavin-T staining assay	[107]
*Calothrix* sp.	Methanolic extract	AChE inhibition	AChE inhibitory assay		[103]
*Tolypothrix* sp.					
*Phormidium* cf. *amoenum*					
*Phormidium* sp.					
*Geitlerinema splendidum*					
*Spirulina* sp.	C-phycocyanin (CPC)	Inhibition of Aβ_40/42_ fibril formation	EM imaging		[108]
		Antioxidant; anti-inflammatory		Wistar rats: bioavailability of C-PC in cortical tissue homogenates; DCFH-DA for ROS levels; Cayman’s protein carbonyl assay; TBARS for lipid peroxidation damage; GPx, GR; GST; SOD; CAT assays; Caspase-12 activation assay; Calpain activation assay; Western Blot for Cox-2-, Nk-kB, IL-6, GAPH, GFAP, NF, MBP, IBA1, CD11b, Nrf2, MT, PGP, Occludin, Claudin, ZO-1, Connexin43, b-actin; Immunohistochemistry for GFAP and DAPI. TUNEL staining	[109]
*Spirulina* sp.	PUFAs:	Reversion of age-related impairments in LTP; spatial learning and depolarization-induced glutamate release; decrease in age-related microglial activation and associated oxidative stress; inhibition of the Aβ-induced LTP; inhibition by EPA	Glia cells: ELISA for IL-1β and IFN_ϒ_ analysis; IL-1β mRNA analysis on agarose gel	Male Wistar rats: assessment of glutamate release by synaptosomal tissue optical density; ROS quantification of hippocampal homogenate (fluorescence); LTP induction and measurement of excitatory postsynaptic potential (EPSP); Young and aged Wistar rats: morris water maze; analysis of LTP; sphyngomielinase; ELISA for 8-hydroxy-2′ deoxyguanosine; immunohistochemical analysis; analysis of fatty acids; Young, middle-aged and aged male Wistar rats: induction of LTPD in performant path-granule cells synapse with electrode; SDS-PAGE for expression of RAGE, CD40, pJNK and PPARϒ on hippocampal homogenate	[118,119,120,121]
*Spirulina* sp.	Water extract	Antioxidant	ABTS assay DPPH assay		[122]
*Spirulina platensis*	C-phycocyanin	Antioxidant	DPPH assay		[123]
	Water extract (SP)	Improve memory function, prevention of Aβ accumulation, reduction of oxidative stress, enhanced catalase activity.		SAMP8 mice: shuttle box: single trial passive avoidance test. Active (shuttle) avoidance test; ambulatory activity in cubic boxes; measurement of Aβ deposition: Immunostaining of sections of the brain with anti-Aβ antibody; redox status: hippocampus, striatum and cortex homogenates separately; lipid peroxidase, SOD, CAT, and GSH-Px activity assays; lipid peroxidation levels by spectrophotometry	[110]
	Oral administration	Reduction of KA-neuronal death in C3 hippocampal cells; antioxidant		Male SW mice treated with kainic acid (KA): determination of the atrophic and nucleolated pyramidal neurons number and volume of observed in the hippocampal region	[111]
*Spirulina maxima*	Methanolic extract	Antioxidant	Brain homogenate incubated with and without extract antioxidant activity by inhibition of peroxidation	Wistar rats: antioxidant activity by lipid peroxidation of liver, plasma, and brain homogenate	[41,114]
	70% ethanolic extract	Suppression of the Aβ-induced toxicity in PC12 cells by decrease oxidative stress, cell death; increase the brain-derived neuro- trophic factor (BDNF) and decrease BACE1	MTT assay; LDH assay; intracellular glutathione; western blot		[115]
	Polyphenolic extracts	Antioxidant	DPPH assay	Male Wistar rats: lipid peroxidation in liver homogenate	[41,116]
	Phycocyanobilin	Neuroprotection by inhibition of NADPH oxidase			[117]
*Nostoc* 78-12A	Nostocarboline	BChE inhibition	AChE inhibitory assay		[104]
*Nostoc ellipsosporum*	Hexane, ethyl acetate and water extract	Antioxidant	Trolox equivalent (TEAC) assay by ABTS radical decolorisation method; Folin Ciocalteu method for total phenolic content		[105]
*Synechococcus* sp.					
*Lyngbya* sp.	Ethanolic fraction	Antioxidant; advanced glycation end-products (AGEs) inhibition	DPPH assay; phosphomolybdenum assay; BSA glycation inhibition assay; nitric oxide inhibition assay for anti-inflammatory activity	*C. elegans* (N2, Bristol) glucose-induced hyperglycemia: HCS analysis for AGE accumulation in live animals; quantitative analysis of AGE accumulation by spectrofluorimetry; DNSA method for glucose analysis; semi-quantitative RT-PCR analysis for stress responses genes (glod-4, daf-16, daf-2.) *C. elegans* TJ356: daf16: GFP:HCS of the localization of daf-16 tagged GFP	[124]
	Tasiamide F	BACE1 inhibition	Antiproteolytic activity		[100]
*Lyngbya majuscula*	Hydroalcoholic extract	Antioxidant and neuroprotective	PC12 cells: MTT assay; DPPH assay; Caspase 3 activity; DNA ladder assay; DAPI staining		[125]
	Kalkitoxin	Blocking agent to Voltage-gated Sodium channels (VGSC)	Cerebellar granule neurons: LDH activity assay Intracellular Ca^2+^ analysis whole cell binding assay		[126]
*Anabaena-flos-aquae*	Anatoxin-a(s)	AChE and BChE inibithion	AChE inhibitory assay	Sprague-Dawley male rats for blood cells: in vitro cholinesterase assays	[102]
	Aquose extract (Klamin^®^)	Anti-inflammatory; protective role against Aβ aggregation	Oxygen Radical Absorvance Capacity (ORAC) assay Folin-ciocalteu Assay LAN5 cells: MTS assay; ROS Generation and Mitochondrial; transmembrane Potential modification through DCFH-DA assay and MitoProbe JC-1 assay kit and MitoSOX Red Reagent; LAN5 cells treated with Aβ oligomers: DCFH-DA assay; MTS assay; Immunostaining for NFkB and Hoescht 33258; ELISA for IL-6 and IL-1β; thioflavin T for Aβ kinetics studies and for the formation and mean size of Aβ aggregates		[41,127]
		Inhibition of H_2_O_2_- induced cytotoxicity and ROS generation; neuroprotection towards Aβ oligomers and Aβ oligomer-induced oxidative stress	Oxygen radical absorbance; Folin-Ciocalteu assay for penolic contents; ABTS assay A549 cells: MTS assay in after no treatment and after H_2_O_2_ exposure; DCFH-DA assay after H_2_O_2_ exposure; LAN5 cells: MTS assay after Aβ treatment oligomers and observation of morphology; DCFH-DA assay after H_2_O_2_ or Aβ oligomers exposure		[128]
*Phormidium* sp. A09DM	Phycoerythrin	Antioxidant	Fibroblast (3T3-L1) cell line: MTT assay; DCFH-DA staining on H_2_O_2_ induced oxidative stress	*Caenorhabditis elegans* N2: DCFH-DA stain: *C. elegans* CL4176 transgenic model of AD: DCFH-DA stain; Heat-induced Paralysis assay. *Drosophila melanogaster*: climbing assay; SOD and CAT assays	[112]
	Allophycocyanin	Improve lifespan, improve rate of survival against oxidative stress and thermal stress; moderate expression of human Aβ1-42 and associated Aβ-induced paralysis		*C. elegans* N2 Bristol (wild type): lifespan study; Stress tolerance assay (H_2_O_2_-induced oxidative stress and thermal stress by increasing temperature from 20 °C to 35 °C); DCHF DA staining; study of lifespan on knockdown skn-1 and daf-16 worms; *C. elegans* TJ356: DAF 16:GFP nuclear localization: evaluating the existence of GFP agglomeration in the nuclei after induce heat shock; *C. elegans* CL4176: paralysis assay after inducement of Aβ1-42 production; Aβ staining by thioflavin T.	[113]
*Trifolium pratense*	Biochanin A	Attenuation of the cytotoxic effect of the Aβ25–35 protein by decreasing viability loss, LDH release, and caspase activity in cells; reduction of cytochrome c and Puma; restoration of Bcl-2/Bax and Bcl-xL/Bax ratio preventing mitochondrial dysfunction	PC12 cells: MTT reduction assay after Aβ exposure; LDH activity; annexin V–FITC and PI; Hoechst 33342; caspase-8 caspase-9 and caspase-3 activity assay; rhodamine 123 fluorescent dye (Rh123); Western blot to Bcl-2, Bcl-xL, Bax, Puma and cytochrome c		[129,130]
Microcystis, Anabaena	Microcystin-LR	Inhibition of Ser/Thr Protein Phosphatases (PPP)	Measurement of phosphatases activities		[131,132]
Nodularia	Nodularin				

The cyanobacteria genus *Lyngbya* has been extensively studied regarding its antioxidant and anti-inflammatory potential. Ethanolic extract from strains of this genus demonstrated an antioxidant activity that exceeded the one from the standard ascorbic acid and a lower IC_50_ value than the standard phloroglucinol (16.42 ± 0.28 µg/mL vs. 52.57 ± 0 µg/mL, respectively), in vitro [124]. These in vitro results were then confirmed in a hyperglycemic *C. elegans* model with live cell imaging by HCS and by the fluorescent quantification of advanced glycation end products (AGEs). AGEs formation in nematodes treated with the ethanolic extract showed a prominent decrease comparing with the non-treated [124]. In addition to the antioxidant studies, the anti-inflammatory activity was also studied through the nitric oxide (NO) radical scavenging assay and by exposing *C. elegans* TJ356 daf16 to *Pseudomonas aeruginosa* [124]. Results revealed a noticeable NO scavenging activity and prevention of the induced inflammatory response of the *C. elegans* treated with the ethanolic extract. The fluorescence intensity of DAF-16 GFP expression was higher in worms treated with the ethanolic extract than in the no-treated worms [124]. Later, the daf-16 gene expression analysis evidenced that the ethanolic extract induces the daf-16 expression in inflammatory worms, which inhibits NF-kB activity and regulates the immune “T helper cells” activation [124].

An hydroalcoholic extract from *Lyngbya majuscula* was studied in the PC12 cell line, and through the activity of the caspase enzyme an inhibition of apoptosis was registered [125]. Also, from a DPPH free radical scavenging activity, it was observed an increased antioxidant potential when compared to ascorbic acid as standard. Also, from *Lyngbya majuscula*, the compound kalkitoxin (Figure 3) was studied for its ability to inhibit tetrodotoxin-sensitive voltage-sensitive sodium channels and it was registered a concentration-dependent inhibition of veratridine-induced elevation of [Ca^2+^]. Through a LDH assay, it was observed that kalkitoxin decreased the veratridine-induced acute neurotoxicity in a concentration-dependent manner [126].

Interesting extracts were also studied for their neuroprotective activity using the neurodegenerative cell model LAN5, treated with Aβ oligomers. According to the cell proliferation MTS assay, the *Aphanizomenon flos-aquae* extract Klamin^®^ supplementation decreased the amyloid-induced toxicity [128]. On these cells Klamin^®^ was not cytotoxic, and it was able to inhibit the TBH-induced toxicity (ROS generation), to counteract TBH induced mitochondrial depolarization and mitochondrial ROS generation, thus, having a highly antioxidant potential. Regarding Aβ toxicity, Klamin^®^ reduced Aβ-induced toxicity and oxidative stress, neuroinflammation (estimated by NF-kB localization) reducing the expression of proinflammatory cytokines. Finally, Klamin^®^ interfered with Aβ aggregation kinetics inducing formation of smaller aggregates [127].

Poly-unsaturated fatty acids (PUFAs) were found to be important for neurological processes, with beneficial effects on memory and learning function, thus, their implementation on dietary could constitute a good therapeutic or preventive approach for AD [118,119,120,121,133]. PUFAs have a limited human endogenous synthesis; however, they have been found to be produced by some cyanobacteria such as *Spirulina plantesis* [134] which highlights the potential of this genus in AD treatment or prevention and opens doors to its study in other genera aimed at this end.

Another interesting compound that might be directed to AD is biochanin (Figure 4). This known phytoestrogen has been identified in cyanobacterial blooms [129]. The pre-treatment with this compound was found to reverse the loss of toxicity induced by Aβ proteins in PC12 cell lines. When PC12 cells were exposed to this peptide, a decrease in caspase activity, LDH release, restoration of Bcl-2/Bax and Bcl-xL/Bax ratio, and reduced expression of cytochrome c and Puma occurred. On this study, the ability to prevent mitochondrial dysfunction was considered as the main reason for increase on cell viability and decrease on apoptosis [130].

Finally, the cyanotoxin microcystin-LR, found in several cyanobacterial genera such as *Anabaena* and *Microcystis*; and nodularin, found on the genus *Nodularia* have the potential to inhibit several ser/thr protein phosphatases (PPP)-family, with a pronounced inhibitory activity on protein phosphatase-2B [131]. These phosphatases have been suggested to play a role in AD tau pathogenic mechanisms towards tau phosphorylation, thus their inhibition through cyanobacterial toxins could be a promising therapeutic [132,135].

As a general overview we present in Table 2 a summary of the main compounds isolated from cyanobacteria and activity that can be directed to AD therapies.

**Table 2 marinedrugs-19-00343-t002:** Summary of the main compounds isolated from cyanobacteria and activity that can be directed to AD therapies.

Compound	Action	Reference
Anatoxin a(s)	AChE and BChE inhibition	[102]
Biochanin	Prevent mitochondria dysfunction	[130]
Kalkitoxin	Inhibition of voltage-gated sodium channels	[126]
Microcystis	Inhibition of Ser/Thr protein phosphatases	[131]
Nodularin	Inhibition of Ser/Thr protein phosphatases	[131]
Nostocarboline	BChE inhibition	[104]
Phycocyanin	BACE1 inhibition; antioxidant	[106]
Phycoerythrin	BACE1 inhibition; antioxidant	[107]
Tasiamide B/F	BACE1 inhibition	[98,100]

## 4. Conclusions

Disease modifying targets to AD are currently attractive, and include targets involved on the Aβ-plaques and NFT’s production, such as the inhibition of BACE1; increase synaptic signaling through the inhibition of AChE; reduction or prevention of oxidative stress and neuroinflammation.

Analyzing current literature on the use of cyanobacteria in AD, cyanobacterial activity is not restricted to specific and isolated compounds; instead, they have shown therapeutic activity in the form of extracts and through diverse routes of administration.

The pigment PC is present in several species of cyanobacteria and several modes of action, including antioxidant and anti-inflammatory activity, inhibition of Aβ40/42 fibril formation, and even BACE1 inhibition were described. Another compound present in more than one genus is PE with both BACE1 inhibition and antioxidant activity.

As referred above, the genus *Spirulina* has relevant antioxidant properties as well as diversity of anti-AD mechanisms, becoming the most studied cyanobacteria genus, which is followed by *Lyngbya* genus. Other cyanobacteria genera require further studies for detection and/or identification of compounds with potential for AD therapies.

There has been some prevalence on the screening for BACE1 and AChE inhibitors despite BACE1 inhibitors failures in clinical and preclinical trials, or AChE inefficiency towards neuroprotection. Antioxidants also occupy a privileged position in terms of investigation, since cyanobacteria have a wide range of antioxidant molecules. BNDF and LTP targeting agents are also being studied among cyanobacteria making additional research essential for the development of this approaches.

Research on the potential of cyanobacteria regarding other promising targets namely in respect to neuroinflammation and to tau-related approaches is lacking and needs to be further explored. Finally, the long-recognized potential of cyanobacteria for AD is still a growing field that requires attention and investment to discover new potential therapeutic agents.

Given the multifactorial nature of AD, further studies with multitargeting approaches combining these new approaches and molecules are logical for developing effective therapies.

## Figures and Tables

**Figure 1 marinedrugs-19-00343-f001:**
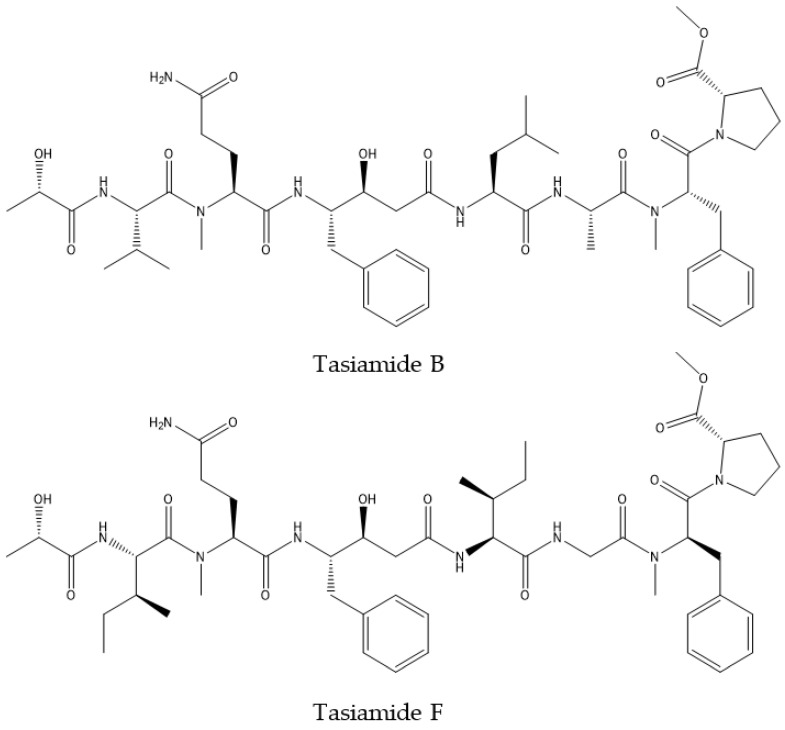
Structure of tasiamide B and its analogue tasiamide F.

**Figure 2 marinedrugs-19-00343-f002:**
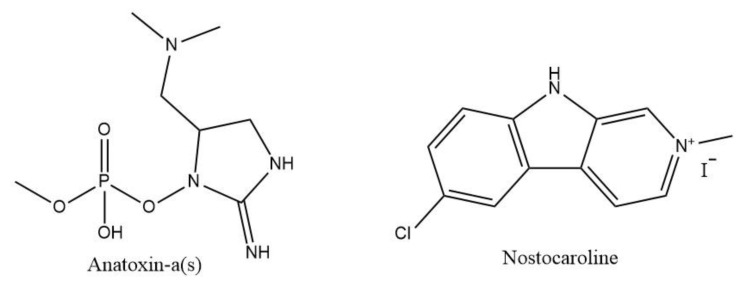
Structure of anatoxin-a(s) and nostocaroline.

**Figure 3 marinedrugs-19-00343-f003:**
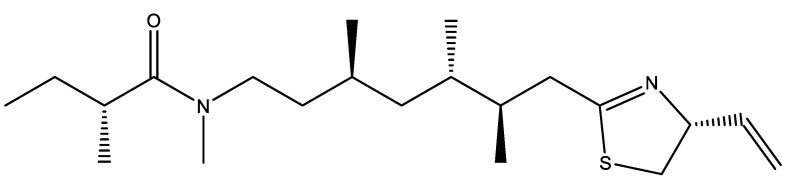
Structure of kalkitoxin.

**Figure 4 marinedrugs-19-00343-f004:**
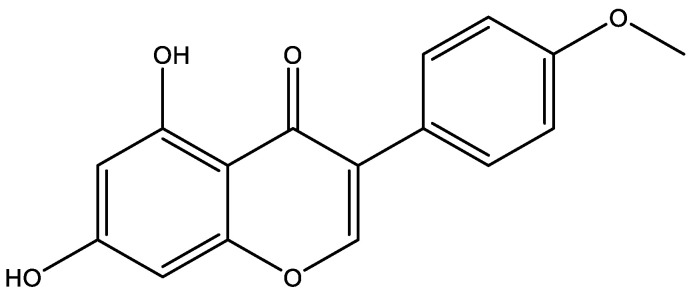
Structure of Biochanin a known phytoestrogen that has been identified in cyanobacterial bloom.

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
