# Peer review of "Bridging Cyanobacteria to Neurodegenerative Diseases: A New Potential Source of Bioactive Compounds against Alzheimer’s Disease"

_marinedrugs, 2021, doi:10.3390/md19060343_

Round 1

Reviewer 1 Report

This review provides a detailed description of Alzheimer's Disease along with a nice description of some of the known targets for the treatment of its symptoms. The review would benefit form the inclusion of the structures of the previously isolated compounds when known (Tasiamides, kalkitoxin, etc.) but otherwise provides a good description of the current state of the field and provides some clear areas where more research is warrented. 

While not needed, if there was anything known about the possible or presumed ecological role of Cholinesterase inhibtiors for cyanobacteria that may be some good additional scholarship that could be added. 

The manuscript is clearly written and provide a good voerview of the current state of the field. 

Reviewer 2 Report

The paper entitled ‘Bridging Cyanobacteria to Neurodegenerative Diseases: a new potential source of bioactive compounds against Alzheimer disease’ aimed to discussion about Alzheimer’s disease, the most important cause of dementia, as well as cyanobacteria revealing promising activity against AD factors and symptoms. Authors presented the most important information about neurodegeneration, pathophysiology, drugs against the disorder as well as cyanobacteria. The paper is well prepared. References are up to date nevertheless some points should be improved:

  1. In my opinion the introduction based on AD is too long (part 2. Alzheimer disease). The presented information are well known and should be presented in short form. An example can be table or short form of proposed description.
  2. If authors decide to short the section 2. Alzheimer disease, AD hypothesis (Alzheimer Disease Hypothesis and main terapheutical targets) should be underlined by subheadings.
  3. The paper should be enriched with figures what makes the paper more readable and attractive

Reviewer 3 Report

Please find attached  file 

Reviewer 4 Report

The present review discusses Cyanobacteria metabolites as new potential sources of bioactive compounds against Alzheimer's disease. The theme is interesting; nevertheless, the authors should proceed to major improvements.

  • The document has many orthographic, grammatical, and syntactical errors that the authors should work with. As an example but not all:
  • Line 90: phicoeritrin
  • Line 91 allophicocianin
  • Line 99: aspartatic protease
  • Line 115: previously
  • Line 117: carers
  • Lines 125 and 126: incidence and Norther regions
  • Line 133: of AD are the presence
  • Line 142: amygadala
  • Line 148: terapheutical targets
  • Line 133: function
  • Line 133: of AD are the presence
  • Line 155: domain
  • Line 156: sulfage
  • Line 256: Glicogen synthetase
  • Line 369: to inibith BACE 1
  • Line 375: of selective BACE1 what?
  • Line 388: showed similar systemic ef fects similar to those observed with anticholinesterase treatments
  • Line 390: Galanthamine or galantamine?
  • Lines 394-395: µlmol Trolox/g, with a total value varying from 2.37+/- 1.15 to 21.09+/- 1.83 µlmol Trolox/g according to the strain [103]
  • Line 404: Moreover
  • Line 409 and elsewhere: neuroprotectant activity Neuroprotective?
  • Line 440: not only reduced the oxidative ROS levels in cell lines and elegans induced
  • Line 445: higther
  • Line 462: maxima phycoyanobilin
  • Line 473: phosphomolybdnenum assay
  • Line 474: pholoroglucinol
  • Line 479: the immune cells “T helper cells” activation
  • Line 518: higthliths
  • Line 521: diretde

In addition, the authors have to work with the document since some sentences are wordy and difficult to understand

As an example:

  • As life expectancy continues to increase, the number of new cases per year is estimated to increase about 7.7 million [32], thus expected to triple to million by 2050, or something like that???
  • Line 272: It has been observed as early events on AD, mitochondrial disfuntion and chronic inflammation [40], being the last caused by inflammatory cytokines release from an excessive deposition of microglial cells [29]. When small Aβ plaques begin to form it has been shown to occur the activation and attraction of microglia observed in AD brain tissue. Please rephrase

In my personal opinion, the documents should be rechecked, probably by a native speaker.

Additionally, I believe that the first part of the document is by far too long. The authors discuss many interesting things regarding AD inhibitors; nevertheless, more than half of the document describes the AD therapeutic targets and some of the most common drugs. Probably the authors should decrease this section

Finally, the authors should add some schemes with the structure of the most important cyanobacteria metabolites with anti-AD activity. These schemes could have a significant impact on the document since they will help the readers (especially the medicinal chemists) to have a quick idea regarding the activity -structure relationship

Round 2

Reviewer 4 Report

The authors have proceeded to significant improvement of the document; thus, in my personal opinion, I believe that this work could be published to Marine drugs

Author Response

Revised manuscript ID: marinedrugs-1242171

Authors’ response to Reviewer 4

Comments and Suggestions for Authors

The authors have proceeded to significant improvement of the document; thus, in my personal opinion, I believe that this work could be published to Marine drugs.

Author response: First, we would like to thank the reviewer's positive feedback to our work. We would like to thank once more for all the corrections, comments and suggestions performed in the first round, which greatly contributed to improve our work.